# GENERATIVE PRE-TRAINED SPEECH LANGUAGE MODEL WITH EFFICIENT HIERARCHICAL TRANSFORMER

## ABSTRACT

While recent advancements in speech language modeling have achieved significant progress, they face remarkable challenges in modelling the long acoustic sequence of neural audio codecs. Previous speech language models are compelled to learn acoustic tokens through a multi-stage generation process, which hinders their performance due to error propagation and information loss. In this paper, we introduce **G**enerative **P**re-Trained **S**peech Language Model (GPST), a hierarchical transformer designed for efficient speech language modeling. GPST quantizes audio waveforms into two distinct types of discrete speech representations and integrates them within a hierarchical transformer architecture, allowing for a unified one-stage generation process and enhancing Hi-Res audio generation capabilities. By training on large corpora of raw audio waveforms in an end-to-end unsupervised manner, GPST can generate syntactically consistent speech with diverse speaker identity unconditionally. When provided a brief 3-second prompt, GPST is able to produce natural and coherent personalized speech, demonstrating in-context learning abilities. Moreover, our approach can be easily extended to spoken cross-lingual speech generation by incorporating multi-lingual semantic tokens and universal acoustic tokens. Experimental results indicate that GPST significantly outperforms the existing speech language models in terms of word error rate, speech quality and speaker similarity.

## 1 INTRODUCTION

Speech quantization has emerged as a crucial technique for speech language models to generate controllable, high-quality speech waveforms (Borsos et al., 2023a; Lakhotia et al., 2021; Wang et al., 2023a; Kreuk et al., 2022; Kharitonov et al., 2023; Borsos et al., 2023b). Specifically, a speech waveform can be quantized into two distinct types of discrete representations: semantic tokens (Lakhotia et al., 2021) and acoustic tokens Défossez et al. (2022); Zeghidour et al. (2021). The semantic tokens are typically obtained by applying the K-means clustering algorithm to the continuous activation space of self-supervised speech models (Hsu et al., 2021; Baevski et al., 2020). Notably, GSLM (Lakhotia et al., 2021) finds that auto-regressive models trained on the semantic tokens can capture high-level linguistic content, supporting language modeling and resynthesis (Polyak et al., 2021). However, semantic tokens fail to retain acoustic details such as speaker identity, resulting in suboptimal reconstruction. In contrast, acoustic tokens generated by neural codec models (Zeghidour et al., 2021; Défossez et al., 2022) effectively compress speech at low bitrates while capturing the nuances of speech waveforms. Consequently, a speech language model can maintain long-term consistency with semantic tokens and produce high-quality synthesis with acoustic tokens.

However, neural codec models require an excessive number of codes for high-quality speech synthesis. For example, EnCodec (Défossez et al., 2022) generates codec embeddings at 75 Hz for audio waveforms at 24 kHz. Subsequently, these codec embeddings are modeled using residual vector quantization (RVQ), wherein high quality synthesis typically requires eight or more hierarchical quantizers with 1024 entries. Therefore, a mere 10-second waveform results in at least $75 \times 8 \times 10 = 6000$ codes, which constitutes an excessively long sequence for language models to learn due to the quadratic complexity with respect to the sequence length for calculating self-

attention (Vaswani et al., 2017). Consequently, addressing the trade-off between perceptual quality and complexity remains a core challenge for speech language models.

Recently, some methods have been proposed to address the issue of long acoustic sequences. The acoustic tokens inherently possess a hierarchical structure because of residual vector quantization: tokens from the preceding quantizers restore acoustic properties such as speaker identity, while the subsequent quantizers capture finer acoustic details. Each quantizer is trained to model the residuals from the previous quantizers. Recent approaches (Borsos et al., 2023a; Wang et al., 2023a; Kharitonov et al., 2023) treat the acoustic token generation process as a multi-stage framework to avoid learning too long sequences. AudioLM (Borsos et al., 2023a) and SPEAR-TTS (Kharitonov et al., 2023) divide acoustic tokens into coarse and fine parts, to which two separate auto-regressive models are applied respectively with semantic tokens as conditions. Although they reduce the sequence lengths that individual models need to handle, they can only generate very short fine speech waveforms. VALL-E (Wang et al., 2023a) uses phonemes as input and acoustic tokens as output for TTS. It conducts auto-regressive generation of the acoustic tokens from the first quantizer and non-auto-regressive generation of the acoustic tokens from subsequent quantizers, which limits the performance of fine acoustic token generation. These multi-stage generative models induce significant error propagation issues, which can negatively impact the overall performance. Additionally, obstructing the information flow among hierarchical quantizers would degrade the model's performance, especially in Hi-Res speech generation that requires more residual quantizers.

In this work, we present **G**enerative **P**re-Trained **S**peech Language Model (GPST), a model that facilitates controllable, high-quality speech generation in a *single stage*. Our approach combines speech quantization with the architecture of a hierarchical transformer (Lee et al., 2022; Yu et al., 2023). Specifically, we follow AudioLM (Borsos et al., 2023a) and discretize raw audio waveforms into semantic tokens and acoustic tokens. We adopt the the semantic extraction model in the multilingual speech translation system SeamlessM4T (Barrault et al., 2023) to support multilingual speech generation. For Hi-Res audio generation, we incorporate the neural codec model EnCodec (Défossez et al., 2022) as the universal acoustic extraction model. The acoustic tokens are represented as a stack of $D$ discrete codes, where $D$ corresponds to the number of residual quantizers in the neural codec model. GPST initially models the semantic sequence with a next token prediction task, followed by modelling the acoustic sequence with the task of predicting the next $D$ stack codes. The semantic sequence serves as a prefix for the acoustic token as a condition. We design a specialized hierarchical architecture to model the underlying hierarchical structure of the acoustic sequence, which comprises of a large global transformer and a small local transformer. The global transformer focuses on learning the high level relationships between the semantic tokens and the stacked acoustic tokens, while the local transformer concentrates on modeling the hierarchical details in the stacked acoustic codes. By incorporating semantic and acoustic tokens within one hierarchical transformer, GPST can significantly reduce the computational costs and effortlessly learn the long-term interactions of semantic tokens and local dependencies among residual codes. Furthermore, we propose a training technique called "local-drop" to further improve the training efficiency for Hi-Res speech generation with a large number of residual quantizers, which is typically impractical in current speech language models. Consequently, our model can generate high-quality and semantically coherent speeches in one stage efficiently.

Our main contributions are summarized as follows.

- We propose a novel generative pre-trained speech language model GPST that enables controllable, high-quality speech generation in a single stage. By integrating semantic tokens and acoustic tokens within a hierarchical transformer, GPST significantly reduces computational costs while efficiently learning the long-term interactions of semantic tokens and local dependencies among residual codes simultaneously.

- We demonstrate GPST's capacity not only in generating coherent speech unconditionally, but also generating speech while preserving the speaker identity with only 3-second short prompt. Experimental results reveal its superiority over existing speech language models with only 33% parameters.

- To the best of our knowledge, GPST is the first work that supports spoken multilingual speech generation and Hi-Res speech synthesis.

## 2 RELATED WORK

### 2.1 DISCRETE SPEECH REPRESENTATION

Speech quantization has become a fundamental technique in speech language modeling (Borsos et al., 2023a; Kreuk et al., 2022; Wang et al., 2023a; Kharitonov et al., 2023). Typically, a speech waveform can be quantized into two distinct types of discrete representations: semantic tokens and acoustic tokens. Benefiting from the development of self-supervised learning in the filed of speech understanding, Textless NLP (Lakhotia et al., 2021; Polyak et al., 2021) proposes to model speech based on HuBERT codes (Hsu et al., 2021) or semantic tokens, which are obtained by applying a K-means clustering algorithm on the activation hidden space of HuBERT. GSLM (Lakhotia et al., 2021) suggests that semantic tokens can capture local dependencies (phonetics) and global long-term structures (language syntax and semantic content). Auto-regressive modeling of these tokens can facilitate generating syntactically and semantically plausible speech continuations. SeamlessM4T (Barrault et al., 2023) learns a spoken multi-lingual SSL model XLSR (Babu et al., 2021) on a large-scale speech dataset to build a multi-lingual semantic vocabulary for speech translation. However, semantic tokens fail to synthesize the acoustic details in speech such as the speaker identity. To address the limitation of suboptimal reconstruction, neural audio codecs (Zeghidour et al., 2021; Défossez et al., 2022) are proposed to quantize speech into stacked codes with residual vector quantization (RVQ) at low bitrates while preserving high-quality reconstruction. These acoustic tokens can capture the details of audio waveforms as diverse as multi-speaker speech (Borsos et al., 2023a), music (Agostinelli et al., 2023) and audio effects (Kreuk et al., 2022). In comparison, the proposed GPST integrates semantic tokens and acoustic tokens within one model in a single stage, effectively unifying the strengths of them.

### 2.2 SPEECH LANGUAGE MODELS

Recently, speech language models have achieved remarkable progress in generating controllable, high-quality speech waveforms. Among them, SpeechGPT (Zhang et al., 2023a) conducts further pre-training and instruction tuning on a speech dataset of semantic tokens, empowering text-based LLMs such as LLaMA (Touvron et al., 2023) to handle cross-modal instruction recognition and speech dialogues. AudioLM (Borsos et al., 2023a) introduces acoustic tokens into semantic token modeling and proposes a multi-stage generative framework to model semantic tokens, coarse acoustic tokens and fine acoustic tokens simultaneously, resulting in semantically coherent and expressive speech generation. SPEAR-TTS (Kharitonov et al., 2023) extends AudioLM to the TTS task by additionally training a text to semantic token model. SoundStrom (Borsos et al., 2023b) speeds up the generation process of AudioLM by introducing confidence-based parallel decoding on acoustic tokens. VALL-E (Wang et al., 2023a) proposes a multi-stage language model for TTS with phonemes as input and acoustic tokens as output. VALL-E X (Zhang et al., 2023b) extends VALL-E to cross-lingual TTS tasks. However, it is based on text-based translation system, failing to handle languages without a written system. Viola (Wang et al., 2023b) proposes a multi-task framework built upon VALL-E to support speech recognition, speech synthesis, and translation tasks. However, they are compelled to model the acoustic tokens in a multi-stage framework due to the high complexity of learning long acoustic sequences. The proposed GPST avoids this limitation by proposing a hierarchical transformer architecture that unifies the semantic tokens and stacked hierarchical acoustic tokens within one stage.

## 3 GENERATIVE PRE-TRAINED SPEECH LANGUAGE MODEL (GPST)

In this section, we start with the formulation of speech language modeling, along with the modeling challenges in speech generation. Next, we elaborate our proposed model GPST in detail, followed by an efficient training technique for GPST to generate Hi-Res speech. Additionally, we discuss various inference modes with in-context learning.

### 3.1 GENERATIVE SPEECH PRE-TRAINING

Given an audio waveform sequence $x \in \mathbb{R}^T$, we quantize it into the sequence of semantic tokens $S = (s_1, \ldots, s_{T_1}) \in \{1, \ldots, N_s\}^{T_1}$ and acoustic tokens $A = (a_1^1, \ldots, a_1^D, \ldots, a_{T_2}^D) \in$

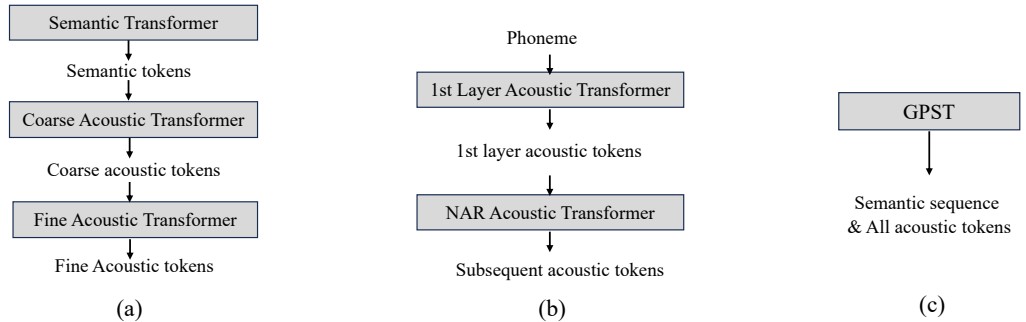

Figure 1: The comparison of frameworks for generative speech pre-training. (a) AudioLM is a three-stage model. (b) VALL-E is a two-stage model. (c) GPST is a one-stage model.

$\{1, \ldots, N_a\}^{T_2 \times D}$, with $T_1, T_2 \ll T$. The acoustic sequence is a two-dimensional matrix and has a hierarchical structure such that $a_t^q$ is derived from the residual of the previous token $a_t^{q-1}$. The learning objective of the speech language model can be auto-regressively factorized as

$$p(S, A) = p(S)p(A|S) = \prod_{t=1}^{T_1} p(s_t|s_{<t}) \prod_{q,t=1}^{D,T_2} p(a_t^q|a_{\leq t}^{<D}, a_t^{<q}, S) \tag{1}$$

A naive approach can unfold the acoustic sequence $A$ into a one-dimensional sequence of length $T_2 \times D$ in raster-scan order and feed it to a transformer model. However, $T_2 \times D$ is typically a large number, and the transformer would suffer from the quadratic cost of its self-attention mechanism.

AudioLM (Borsos et al., 2023a) adopts a three-stage approach for modeling speech, as depicted in Figure 1(a). The first stage involves auto-regressive pre-training on semantic tokens to capture the long-term temporal structure. Next, the acoustic sequence, which is of size $T_2 \times D$, is divided into a coarse part of size $T_2 \times D'$ and a fine part of size $T_3 \times (D - D')$, where $D'$ is typically much smaller than $D - D'$. The fine part is a small subset of the coarse sequence to reduce the sequence length since $T_2 \times (D - D')$ is still too large. AudioLM designs two individual transformers to model the coarse and fine acoustic sequences separately. The learning objective is *approximately* factorized as

$$p(S, A) = p(S)p(A|S)$$
$$\approx \prod_{t=1}^{T_1} p(s_t|s_{<t}; \theta_S) \prod_{q_1, t=1}^{D', T_2} p(a_t^{q_1}|a_{\leq t}^{<D'}, a_t^{<q_1}, S; \theta_C) \prod_{q_2=D'+1, t=1}^{D, T_3} p(a_t^{q_2}|a_{<t}^{<q_2}, a_t^{<q_2}, a_{\leq T_3}^{\leq D'}; \theta_F) \tag{2}$$

where $q_1 \leq D' < q_2 \leq D$ and $T_3 < T_2$. The fine acoustic transformer only models a small subset of the coarse acoustic tokens to reduce the sequence length. The learnable parameters $\theta_S, \theta_C, \theta_F$ correspond to three independent transformers respectively.

As shown in Figure 1(b), VALL-E (Wang et al., 2023a) uses phoneme sequences derived from text with a G2P tool as the condition, rather than semantic tokens. We slightly abuse the notation here since phonemes serve a similar purpose with semantic tokens. VALL-E also divides the acoustic token generation process into two stages, where the acoustic tokens from first quantizer layer are generated in an auto-regressive manner while the subsequent acoustic tokens are generated in non-auto-regressively. Note that VALL-E can not generate semantically coherent sequences unconditionally since it does not model $p(S)$. The learning objective is *approximately* factorized as

$$p(A|S) = \prod_{q,t=1}^{D,T_2} p(a_t^q|a_{\leq t}^{<D}, a_t^{<q}, S) \approx \prod_{t=1}^{T_2} p(a_t^1|a_{<t}^1, S; \theta_{AR}) \prod_{q=2, t=1}^{D,T_2} p(a_t^q|a_{\leq T_2}^{<q}, S; \theta_{NAR}) \tag{3}$$

where $\theta_{AR}, \theta_{NAR}$ refer to different models respectively.

The speech language models above are necessitated to split the acoustic generation into a multi-stage process due to the considerable length of acoustic sequences.

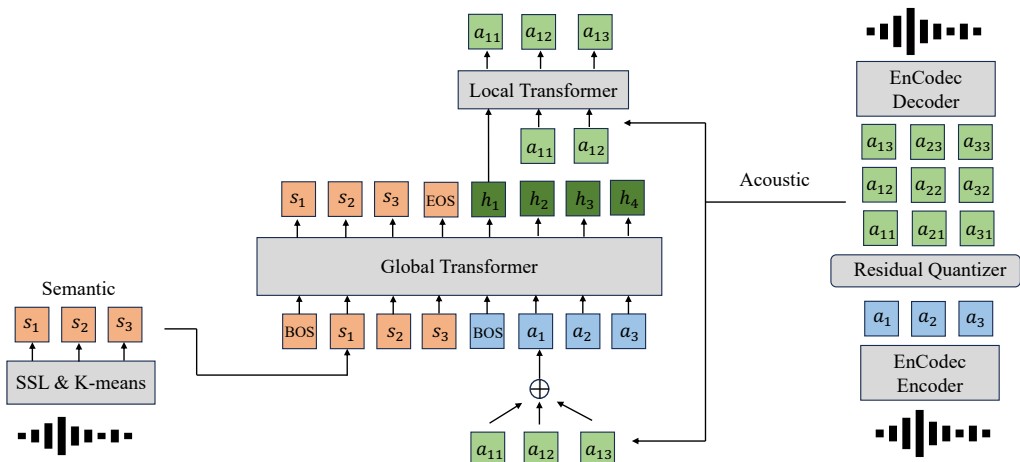

Figure 2: An overview of the framework. The framework is composed of three components: (1) The semantic token extractor with a speech SSL model and K-means for speech discretization. (2) The acoustic token extractor with the neural codec model for speech discretization. (3) The proposed GPST model, which is composed of a global transformer and a local transformer.

## 3.2 EFFICIENT HIERARCHICAL TRANSFORMER

Considering the hierarchical structure underlying acoustic sequence, we propose GPST, a hierarchical transformer architecture to effectively and efficiently learn the codes extracted by EnCodec. As shown in Figure 2, GPST is composed of (1) a semantic token extractor that integrates a speech SSL encoder and a K-means clustering model (Barrault et al., 2023), as well as a neural codec model EnCodec (Défossez et al., 2022), (2) a large global transformer that contextualizes representations by applying self-attention over previous semantic tokens and stacked acoustic tokens, and (3) a smaller local transformer that takes a contextualized hidden state from the global model, and auto-regressively predicts subsequent acoustic codes. We adopt the setting of a large global module with a small local module to simulate potential applications that use LLMs as the global module, which we leave for future work. The learning objective is *exactly* factorized as

$$p(S, A) = p(S)p(A|S) = \prod_{t=1}^{T_1} p(s_t|s_{<t}; \theta_{global}) \prod_{q,t=1}^{D,T_2} p(a_t^q|a_{<t}^{\leq D}, a_t^{<q}, S; \theta_{global}, \theta_{local}) \quad (4)$$

The end-to-end learning process is within one model in one stage, which avoids information loss and mitigates the significant error propagation issues that can arise in a multi-stage formulation.

**Global Transformer.** The global transformer is an $N_g$ layer decoder-only transformer with a causal mask. It has two types of tokens concatenated into a single sequence as input. The first type comprises semantic tokens, which can capture long-term consistency. The second type is the sum of the acoustic tokens obtained by RVQ

$$E(s_t) = E_s(s_t) + \text{PE}_g(t), \text{for } 1 \leq t \leq T_1$$

$$E(a_t) = \sum_{q=1}^{D} E_a(a_t^q) + \text{PE}_g(t + T_1), \text{for } 1 \leq t \leq T_2 \quad (5)$$

$$h_t = \text{GlobalTransformer}(s_1, \ldots, s_{T_1}, a_1, \ldots, a_{T_2}), 1 \leq t \leq T_1 + T_2$$

where $E_s$ and $E_a$ are embedding functions for semantic and acoustic tokens respectively. $\text{PE}_g$ is a positional embedding for the global transformer. We add special tokens at the first position and the segment boundary of the sequence to inform the model to switch the generation space.

**Local Transformer.** The local transformer consists of $N_l$ layers. Given the contextualized hidden states $h_t$ from the global transformer, the local transformer auto-regressively predicts $D$ acoustic

codes $a_t^1, \ldots, a_t^D$ at position $t$

$$E(a_t^q) = E_a(a_t^q) + \text{PE}_l(q), 1 \leq q \leq D$$
$$a_t = \text{LocalTransformer}(h_t, a_t^1, \ldots, a_t^D) \tag{6}$$

where $\text{PE}_l$ is a positional embedding for the local transformer, which is shared across position $t$. GPST is trained to minimize the negative log-likelihood:

$$\mathcal{L} = \sum_{t=1}^{T_1} -\log p(s_t | s_{<t}; \theta_{global}) - \sum_{t=1}^{T_2} \sum_{q=1}^{D} \log p(a_t^q | a_t^{<q}, S; \theta_{global}, \theta_{local}) \tag{7}$$

**Local-drop.** The number of residual quantizers increases when generating Hi-Res speech, which would cause high computation complexity. Since the local transformer only models individual stacks of acoustic tokens, it has an input shape of $(\text{Batch} \times T_2, D)$. The dimension of acoustic sequence length $T_2$ is unfolded to the first batch dimension, which means the stack of codes are not attended by self-attention. We propose a technique named local-drop to further improve the training efficiency of GPST. We randomly drop some tokens $a_{\hat{t}}^{\leq D}$ to decrease the size of the first dimension.

### 3.3 INFERENCE

Speech language models can generate semantically coherent speech for unseen speakers with in-context learning, which is an emerging capability of auto-regressive pre-trained language models like GPT (Brown et al., 2020) for zero shot learning. Suppose we have the semantic tokens $S_p$ and the acoustic tokens $A_p$ from the prompt, the semantic tokens $S_t$ and the acoustic tokens $A_t$ from the target. Based on the usage of the prompt, we can categorize the generation mode into four cases.

**Unconditional Generation.** In this setting, we unconditionally generate the semantic tokens, which are subsequently used as the condition for acoustic generation. The randomly sampled semantic sequence can generate diverse, syntactically and semantically consistent linguistic content. The acoustic tokens vary in speaker identity, prosody with the semantic content serving as a guideline. We provide some transcription cases in Appendix A.1.

**Semantic to Acoustic.** In this setting, we use the ground-truth semantic tokens $S_t$ as condition for acoustic generation, which is similar to the task of TTS. The generated speech preserves the content of the spoken sentence while varying in speaker identity. We also follow SPEAR-TTS (Kharitonov et al., 2023) and train a toy decoder-only transformer named GPST-TTS on the LibriSpeech 960h dataset to generate semantic tokens with text as condition, supporting the TTS task.

**Speaker Identity Transfer.** In this setting, we are interested in the task of voice conversion that transfers the speaker identity of the prompt speech into the target speech. The sequence input to the model is concatenated in the following order $[S_p, S_t, A_p]$. GPST is encouraged to generate subsequent acoustic tokens that share the speaker identity with $A_p$ while remaining consistent with the content of $S_t$. We find that directly concatenating linguistically inconsistent $S_p$ and $S_t$ together would cause unstable generation around the interface boundary. To address this issue, we propose artificially inserting a very short silence excerpt (0.1 second) between $S_p$ and $S_t$ to explicitly break the linguistic continuation. In this way, the model would not struggle to mitigate the discontinuity of $S_p$ and $S_t$ and is able to generate stable speeches.

**Acoustic Continuations.** Different from the speaker identity transfer mode that the prompt and target are from different utterance, the prompt of the acoustic continuations mode is the first 3 seconds of the target. The model is asked to generate the acoustic continuations after 3 seconds.

### 3.4 SPOKEN MULTILINGUAL LEARNING

We adopt the multi-lingual XLSR encoder from SeamlessM4T (Barrault et al., 2023) as the semantic token extractor. The semantic vocabulary of SeamlessM4T naturally supports the multi-lingual speech representation. For acoustic tokens, we adopt the pre-trained neural audio codec model EnCodec (Défossez et al., 2022) as the acoustic token extractor. Although EnCodec is trained on the English data, we find that it can synthesize other languages as well. We take it as a universal acoustic extractor.

## 3.5 Efficiency Analysis

Transformer (Vaswani et al., 2017) is criticized for the quadratic complexity with respect to sequence lengths during self-attention calculations. Considering an acoustic matrix $A$ of size $T_2 \times D$, the naive approach of unfolding it into a one-dimensional sequence like AudioLM would result in a computational complexity of $O(NT_2^2D^2)$, where $N$ is the number of transformer layers. In contrast, GPST has $N_g$ global layers and $N_l$ local layers, with the global transformer dealing with a sequence length of $T_2$ and the local transformer with a sequence length of $D$. Suppose $N = N_g + N_l$ for simplicity. The overall computational complexity for GPST is $O(N_gT_2^2 + N_lT_2D^2)$, which is smaller than $O(NT_2^2D^2)$. Furthermore, self-attention is not the main computational cost factor of large transformers. The embedding size and the dimension of the feedforward network dominate the model's overall computational cost (Kaplan et al., 2020). A forward pass with a large transformer with $m$ non-embedding parameters on a sequence of length $T_2$ uses roughly $2mT_2$ FLOPS. Therefore, for GPST with a global dimension $m_g$ and a local dimension $m_l$, the required FLOPS is $2T_2(m_g + m_lD)$. Since $m_l$ is typically much smaller than $m_g$, the FLOPS for GPST is approximately $2T_2m_g$, which is $D$ times faster than the normal transformer with $2T_2Dm_g$ FLOPS.

# 4 Experiments

## 4.1 Experiment Setup

### 4.1.1 Datasets

We follow Borsos et al. (2023a) and use LibriLight (Kahn et al., 2020) as the training data which contains 60K hours of unlabelled speech in English. We randomly crop 10 seconds out of each audio clip for training. We choose LibriSpeech test-clean dataset (Panayotov et al., 2015) for evaluation since there is no speaker overlap with LibriLight. Following Borsos et al. (2023a); Wang et al. (2023a), we select the samples with lengths between 4 and 10 seconds as the test dataset. For the multi-lingual task, we test in a bi-lingual setting with the tone language Chinese and non-tone language English for simplicity. We choose LibriSpeech 960h as the English training data and Aishell-2 1000h (Du et al., 2018) as the Chinese training data, both of which share similar sizes. All experiments are conducted three times and the average scores are reported.

We leverage the XLSR v2 (Babu et al., 2021) model released by SeamlessM4T (Barrault et al., 2023) to extract semantic tokens, resulting in a rate of 50 tokens per second. We remove the consecutive duplicate semantic tokens since such duplicates would cause the generation failures (Lakhotia et al., 2021). We adopt the neural audio codec model EnCodec (Défossez et al., 2022) to extract acoustic tokens, which produce codes at 75 Hz. We choose 8 hierarchical quantizers as the default setting as VALL-E, leading to $75 \times 8 = 600$ tokens per second. We also test a larger bitrate setting for GPST-Hi-Res with 16 quantizers, which is not applicable to other baselines.

### 4.1.2 Implementation Details

Each layer of the global transformer in GPST has 16 attention heads, an embedding size of 1024 with a feed-forward layer dimension of 4096. Each layer of the local transformer is smaller than the global transformer, with 8 attention heads, an embedding size of 512, and a feed-forward layer dimension of 2048. We set the probability of a local drop to $0.5$ only for Hi-Res generation. We provide more training details in Appendix A.2.

### 4.1.3 Baselines

We choose speech language models GSLM (Lakhotia et al., 2021), AudioLM (Borsos et al., 2023a) and VALL-E Wang et al. (2023a) as baselines, together with YourTTS (Casanova et al., 2022) as the TTS baseline. We notice that SoundStorm (Borsos et al., 2023b) improves the multi-stage acoustic generation. However, SoundStorm takes duplicate semantic tokens as the condition, which is an inappropriate setting for other baselines since all the other models remove the consecutive repetitions, and duplicate semantic tokens would reduce the difficulty of acoustic generation. Also, duplicate semantic tokens would cause failures in the generation of semantic tokens (Lakhotia et al., 2021)

Table 1: Evaluation results of speech generation on LibriSpeech test-clean dataset. The WER result of AudioLM is obtained by a Conformer Transducer model, while others are obatined by HuBERT-Large finetuned on LibriSpeech 960h. AudioLM and SpearTTS use the neural codec model Sound-Stream (Zeghidour et al., 2021) while VALL-E and GPST use Encodec (Défossez et al., 2022).

| Model | WER ↓ | SPK ↑ | DNSMOS↑ | # codec | # params |
|---|---|---|---|---|---|
| GroundTruth | 2.2 | 0.754 | - | | |
| **Semantic to Acoustic** | | | | | |
| GSLM (Lakhotia et al., 2021) | 12.4 | - | - | - | - |
| AudioLM* (Borsos et al., 2023a) | 6.0 | - | - | 12 | 300M+300M |
| GPST-TTS (ours) | 4.3 | - | - | 8 | 182M+190M |
| GPST (ours) | **4.0** | - | - | 8 | 190M |
| GPST-Hi-Res (ours) | 6.4 | - | - | 16 | 207M |
| **Speaker Identity Transfer** | | | | | |
| YourTTS (Casanova et al., 2022) | 7.7 | 0.337 | - | - | - |
| AudioLM (Borsos et al., 2023a) | - | 0.460 | - | 12 | 300M+300M |
| SPEAR-TTS (Kharitonov et al., 2023) | - | 0.560 | 3.68 | 3 | 97M |
| VALL-E (Wang et al., 2023a) | 5.9 | 0.580 | 3.87 | 8 | 165M+172M |
| GPST (ours) | **4.2** | **0.605** | 3.89 | 8 | 190M |
| GPST-Hi-Res (ours) | 5.3 | 0.587 | **4.02** | 16 | 207M |
| **Acoustic Continuations** | | | | | |
| VALL-E (Wang et al., 2023a) | 3.8 | 0.508 | - | 8 | 165M+172M |
| GPST (ours) | **2.8** | **0.536** | - | 8 | 190M |
| GPST-Hi-Res (ours) | 3.5 | 0.529 | - | 16 | 207M |

Table 2: Evaluation results on multi-lingual datasets. We use WER for En and CER for Zh.

| | WER/CER | SPK |
|---|---|---|
| Zh-GroundTruth | 26.4 | 0.453 |
| En | 4.1 | 0.501 |
| Zh | 30.2 | 0.430 |
| **Zero-Shot Corss-Lingual Transfer** | | |
| Zh | 33.3 | 0.417 |

Table 3: Ablation study of the model architecture. The model is tested in Acoustic Continuations inference mode.

| $N_g + N_l$ | WER ↓ | SPK ↑ | # params |
|---|---|---|---|
| 11 + 4 | 3.2 | 0.531 | 190M |
| 10 + 8 | 3.1 | 0.532 | 190M |
| 9 + 12 | 2.8 | 0.536 | 190M |

that limits the application in speech generation Kharitonov et al. (2023) and resynthesis for speech translation system (Lee et al., 2021). Therefore, we do not take SoundStorm for comparison here.

### 4.1.4 EVALUATION METRICS

The synthesized speech should align with the semantic input and match the voice of the prompt. Therefore, we are interested in the following metrics for speech language models: (1) word error rate (WER), (2) speaker similarity (SPK), and (3) speech quality (DNSMOS). We employ the HuBERT-Large (Hsu et al., 2021) model as the ASR model for English to calculate WER and Wav2Vec2-XLSR-53 (Baevski et al., 2020) for Chinese to calculate CER. We take the publicly available speaker verification model WavLM-TDNN (Chen et al., 2022) to evaluate the speaker similarity between the prompt and the synthesized speech. We use a MOS estimator DNSMOS (Reddy et al., 2021) to estimate the perceived audio quality of the generated samples. Since the baselines are not open-sourced, we compare DNSMOS with the examples provided on VALL-E's demo page for fairness. Appendix A.3 lists all the evaluation tools.

### 4.2 RESULTS AND ANALYSIS

**LibriSpeech Evaluation.** Table 1 summarizes the results of different inference modes. Compared to the baseline models, GPST achieves the best results in terms of WER, SPK, and DNSMOS. In the semantic to acoustic mode, GPST reaches the lowest WER score with only 33% parameters of

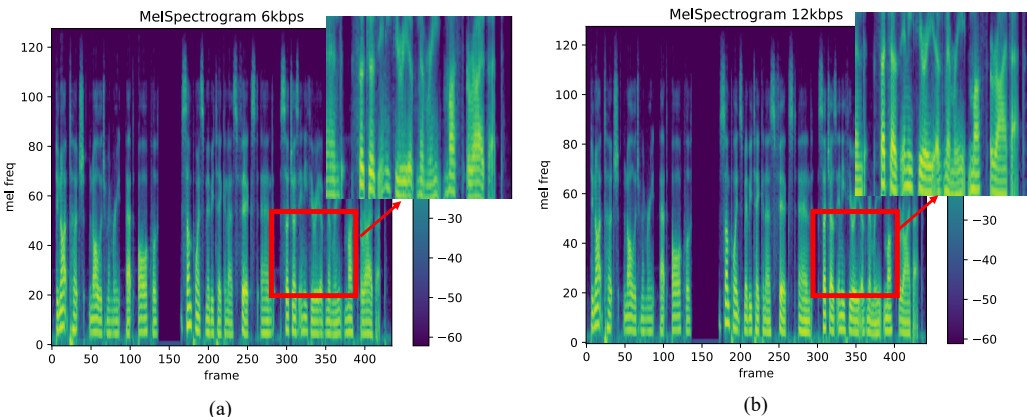

Figure 3: The comparison of mel-spectrograms generated by GPST with (a) 6kbps and (b) 12kbps (Hi-Res). The harmonic energy in the high-frequency of 12kbps is richer.

AudioLM. The quality of semantic tokens is constrained due to the use of a toy model for text to semantic generation, resulting in a minor performance drop of GPST-TTS. We expect that more training data would further improve the TTS performance. We also notice a performance drop in GPST-Hi-Res, which indicates that Hi-Res speech generation, with more quantizers, is still a tough task for speech language models. In the speaker identity transfer mode, GPST achieves the best scores in all the metrics, validating that GPST can better transfer the speaker identity while maintaining the spoken content. It is worth noting that GPST-Hi-Res gets better DNSMOS than GPST, largely because more quantizers can preserve more acoustic details.

**Multilingual Evaluation.** Table 2 shows the results of GPST on multi-lingual datasets. Although trained on a small dataset, GPST demonstrates its generalization ability in multi-lingual settings. Since the Aishell-2 Chinese dataset is noisy, the CER score is low even for the GroundTruth. However, GPST can still achieve a score close to the GroundTruth, which proves the robustness of the model. We also design a Zero-Shot Cross-Lingual Transfer for Multi-lingual settings. We adopt the model trained on English LibriLight only, while inference is conducted on Chinese Aishell-2 with Acoustic Continuations mode without any further training. GPST shows the performance close to the model especially trained on Chinese Aishell-2, which demonstrates GSPT's support for spoken multi-lingual tasks and its strong in-context learning capability of speech language models.

**Effect of Model Architecture Settings:** We conduct an ablation study on the number of layers for the global and local transformer. To match the parameters of every stage in AudioLM or VALL-E for comparison, both of which consist of a global transformer with 12 layers, we adjust the total parameters of the global transformer and local transformer in GSPT to be approximately equal. Since the parameters of one global transformer layer equals four local transformer layers, we adopt the setting of $(12 - x) \times N_g + 4 \times x \times N_l$, where $x \in [1, 11]$. Table 3 shows that increasing the layer number of the local transformer helps GPST learn acoustic tokens better, further improving the performance of acoustic generation.

**On the Hi-Res Quality.** We plot the mel-spectrograms of the same speech generated by GPST with 6kbps (8 quantizers) and 12kbps (16 quantizers) respectively in Figure 3. Generally, richer harmonic energy in the high-frequency regions indicates higher speech quality. As observed, the speech generated by GPST with more quantizers exhibits more details in the mel-spectrogram.

## 5 CONCLUSION

We introduce GPST, a generative pre-trained speech language model that integrates semantic tokens and acoustic tokens within a hierarchical transformer architecture, allowing for a unified one-stage generation process. GPST demonstrates its capability to generate coherent speech and speaker identity transfer with in-context learning. Furthermore, we show that GPST can generate Hi-Res speech and spoken multi-lingual speech as well.

## 6 REPRODUCIBILITY AND ETHICS STATEMENT

To ensure the work is as reproducible as possible, all the tools we adopt in the paper are open-sourced for easy reproduction and the links are listed in Appendix A.3. We have described the implementation details in Section 4.1 and Appendix A.2.

GPST can be directly attached to the Multilingual Multimodal Machine Translation system (Barrault et al., 2023), empowering the system to generate human-like and personalized speech with a brief 3-second sample speech. However, GPST also presents new risks, such as the potential for malicious actors to impersonate public figures or commit fraud. To mitigate such risks, it is possible to watermark the generated speech that is invisible to humans but algorithmically detectable.

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

## A   APPENDIX

### A.1   UNCONDITIONAL GENERATION CASES

Table 4: Transcriptions of some unconditional generation samples.

| |
|---|
| SO WE ARRIVED DRIVING ON FURTHER BUT THEN THE WORSE PRESENTS RECEIVED HIM TO BED END OF CHAPTER SEVENTEEN THE RECORDING BY GRACE SANDERS |
| SPEECH OF THE PRESIDENT IS WITHOUT DIFFICULTY AND WITHDRAWAL AND FROM THAT DEATH OF THE OFFICER HIS KING SAYS BEFORE HE WENT UNTO THE PAPERS AND THE |
| IS FAIR IN THE BACK ROOM AND BETTER WIND TO THE FARTHER SEA THAN THIS BUT STILL AS TO THE SEA SHE FELT HIM IN CRY AND THEN SAID THAT MAN COMING |
| TO THE SAME SOULS AS TO STAND ONWARDS WE SAW HER SUNSET FORTH TO OUR HANDS TOGETHER WITH ONE ANOTHER THE TALES OF PRAYER AND TALENTS INSTINCTIVE |
| THE SIZE OF THE BRANCH OF THE WINTER UNTIL IT WAS TOLD THAT MOSES CALLED POULTRY CORPORATION TO THE FEMALE SO THAT ALL THE SAVAGES ENBODIES IN THE BODIES OR IN THE BLISS IF |

### A.2   IMPLEMENTATION DETAILS

The models are trained on LibriLight using 16 NVIDIA TESLA V100 32GB GPUs with a batch size of 64 for 1M steps, which takes about 1 week. The multi-lingual models are trained for 400K steps. We use the Adam optimizer with a learning rate of 0.0005 and an inverse square root learning rate decay schedule with 10K warm-up steps. To prevent over-fitting, we use label smoothing of 0.1 for training.

### A.3   OPEN-SOURCED TOOLS

ASR HuBERT-Large: `https://github.com/facebookresearch/fairseq/tree/main/examples/hubert`

ASR Wav2Vec2-Large-XLSR-53-Chinese-zh-cn-gpt: `https://huggingface.co/ydshieh/wav2vec2-large-xlsr-53-chinese-zh-cn-gpt`

Speaker verification WavLM-TDNN: `https://github.com/microsoft/UniSpeech/tree/main/downstreams/speaker_verification`

DNSMOS: `https://github.com/microsoft/DNS-Challenge/tree/master/DNSMOS`

VALL-E demo page more samples: `https://www.microsoft.com/en-us/research/project/vall-e-x/vall-e`

SeamlessM4T: `https://github.com/facebookresearch/seamless_communication`

EnCodec: `https://github.com/facebookresearch/encodec`

