# OpenReview forum: "Generative Pre-Trained Speech Language Model with Efficient Hierarchical Transformer"
_ICLR.cc/2024/Conference — Submitted to ICLR 2024_

### Official Review · Reviewer_tzty · 2023-10-28

**Soundness:** 3 good
**Presentation:** 3 good
**Contribution:** 2 fair
**Rating:** 5
**Confidence:** 4

**Summary:**

The paper introduces a model called Generative Pre-Trained Speech Language Model (GPST) that addresses the challenges of modeling long acoustic sequences in neural audio codecs. Existing speech language models struggle to accurately represent acoustic tokens and often suffer from error propagation and information loss. GPST overcomes these limitations by employing a hierarchical transformer architecture and quantizing audio waveforms into two discrete speech representations. This allows for a unified one-stage generation process and improves the generation of high-resolution audio.

**Strengths:**

The main contribution of the paper is the proposal of a unified stage for speech generative models, which contrasts with prior works that typically adopt two or even three stages. One of the main challenges of one-stage modeling is the increased GPU memory and time cost associated with the unified structure. Compared to two-stage models, the prior unified AR approach can result in a significant increase, up to 8 or even 16 times, in cost. To address this problem, the authors employ a relatively small Transformer model, where the top layers are not as large as the bottom layers. Then the memory and time cost are significantly reduced.

The authors employed a few of objective metrics to show the effectiveness of the proposed method.

**Weaknesses:**

1. The primary limitation of the paper is the insufficient evaluation. Due to the absence of a perfect evaluation metric for speech generation tasks, it is necessary to conduct a subjective evaluation in the experiments. Additionally, it would be advantageous to provide a demo page for the reviewers to experience the quality of the pre-trained model firsthand. Even if the authors try their best to do objective evaluations, subjective evaluation is still required. DNSMOS is not enough to measure the speech quality.

2. The primary focus of the paper is on efficiency; however, no specific metric is provided to evaluate its efficiency. I am curious about the potential inference time savings that can be achieved with the HIERARCHICALTRANSFORMER architecture.

3. Given the existence of previous full NAR approaches like SoundStorm, the contribution of the paper seems relatively limited. The efficiency issue has already been largely addressed by these full NAR approaches. Consequently, I have doubts about whether the proposed HIERARCHICALTRANSFORMER structure will inspire further advancements in speech pre-trained models.

4.The storytelling in the paper could be further enhanced. The primary contribution of the paper is the introduction of a unified stage speech pre-trained model, rather than the efficiency of the hierarchical transformer structure. The title and writing of the paper may be slightly misleading for readers.

**Questions:**

1. I am curious about the training and inference time of the structure.

2. Since the model directly predicts all encodec tokens, is there any memory issue in long sequence prediction?

---

> ### Author Response · Authors · 2023-11-15
> **Response part 1/2**
>
> We sincerely appreciate the valuable comments and constructive suggestions to help improve the paper. We give the response below.
>
> ## Q1: The evaluation with subjective metrics and the demo page. DNSMOS is not enough to measure the speech quality.
>
> We do want to conduct the subjective comparison such as MOS for all the models, but sadly AudioLM and VALL-E are not open-sourced, we cannot perform a fair and rigorous test of MOS. We have tried our best to conduct fair and rigorous experiments to compare the baselines with our model, which shares the same protocol for all the models. DNSMOS metric is widely accepted in SPEAR-TTS, VALL-E, and SoundStorm, we believe it can reflect the quality of the audio to some extent. We can provide some cases on the anonymous demo page (https://gpsticlr.github.io/GPST/demo) and the code will be available too.
>
> ## Q2: The question about efficiency and storytelling.
>
> We must clarify that the main contribution of our work is a unified pre-trained speech language model that supports Hi-Res speech generation and spoken multi-lingual. And the efficient hierarchical transformer is the approach to achieve it. What the reviewer states in Q4 also agrees with it. However, there appear to be conflicting assertions in Q2 and Q3, suggesting that the primary emphasis of our work lies in efficiency, which doesn't accurately reflect our main focus. We provide our generation speed test with the fairseq-generate tool. The inference is run on the NVIDIA 3090 GPU in the semantic to acoustic mode with a batch size of 64. However, we cannot provide the speed test results for other baselines since both AudioLM and VALL-E are not open-source.
>
>  |Model | $N_g + N_l$ | speed | # codec |# params|
> |-|-|-|-|-|
>  |GPST | 11+4 | 2.31 sentences/s, 9438.75 tokens/s | 8 | 190M |
>  |GPST | 10+8 | 1.89 sentences/s, 7643.03 tokens/s | 8 | 190M |
>  |GPST | 9+12 | 1.57 sentences/s, 6232.50 tokens/s | 8 | 190M |
>  |GPST-Hi-Res | 9+12 | 0.88 sentences/s, 7186.19 tokens/s | 16 | 207M |
>
> ## Q3: The potential ability of auto-regressive hierarchical transformer and the shortcomings of the NAR model SoundStorm.
>
> We emphasize that efficiency is NOT the primary focus of our model GPST, instead, the main contribution is the unified speech LM framework. The design of GPST considers the potential possibility of integrating the speech generation ability into auto-regressive LLMs like LLaMA, which is not possible for NAR models. For the model architecture design, we do not blindly pursue the number of parameters for GPST, instead, we carefully configure the experimental setting of a large global model and a small local model, simulating scenarios where a small model integrates with LLMs.
>
> We provide the reason why SoundStorm is not appropriate for comparison. It utilizes duplicate semantic tokens as a condition to generate acoustic tokens, leading to a leakage of duration information. Moreover, duplicate semantic tokens would cause failures in the generation of semantic tokens [1], which would limit the application in speech generation [2] and resynthesis for speech translation system [3], while our model GPST can easily assimilate into the SOTA multi-lingual speech translation system SeamlessM4T [4]. Although NAR approach such as SoundStorm can accelerate the generation speed, it exhibit performance limitations when compared to AR models [5,6,7,8]. According to their experiment results, the SPK result for SoundStorm is $0.57$ while GPST achieves a better result of $0.605$.

---

> ### Author Response · Authors · 2023-11-15
> **Response part 2/2**
>
> ## Q4: The memory issue in long sequence generation.
>
> The memory issue in long sequence generation is one of the motivations for GPST as well. The hierarchical structure effectively reduces the sequence length handled by the global transformer, mitigating the memory pressure with lengthy sequences. By allowing the global transformer to concentrate on high-level or coarse features, GPST delegates fine-grained details to local transformers. In our experiments, GPST notably did not encounter memory issues. The low WER indicates the model's competence in handling lengthy sequences without experiencing memory-related challenges.
>
>
> [1] Kushal Lakhotia, Eugene Kharitonov, Wei-Ning Hsu, Yossi Adi, Adam Polyak, Benjamin Bolte, Tu-Anh Nguyen, Jade Copet, Alexei Baevski, Abdelrahman Mohamed, et al. On generative spoken language modeling from raw audio. Transactions of the Association for Computational Linguistics, 9:1336–1354, 2021.
>
> [2] Eugene Kharitonov, Damien Vincent, Zala'n Borsos, Raphae'l Marinier, Sertan Girgin, Olivier Pietquin, Matt Sharifi, Marco Tagliasacchi, and Neil Zeghidour. Speak, read and prompt: High-fidelity text-to-speech with minimal supervision. arXiv preprint arXiv:2302.03540, 2023.
>
> [3] Ann Lee, Peng-Jen Chen, Changhan Wang, Jiatao Gu, Sravya Popuri, Xutai Ma, Adam Polyak, Yossi Adi, Qing He, Yun Tang, et al. Direct speech-to-speech translation with discrete units. arXiv preprint arXiv:2107.05604, 2021.
>
> [4] Lo'ıc Barrault, Yu-An Chung, Mariano Cora Meglioli, David Dale, Ning Dong, Paul-Ambroise Duquenne, Hady Elsahar, Hongyu Gong, Kevin Heffernan, John Hoffman, et al. Seamlessm4t- massively multilingual \& multimodal machine translation. arXiv preprint arXiv:2308.11596, 2023.
>
> [5] Gu, J., Bradbury, J., Xiong, C., Li, V.O. and Socher, R., 2017. Non-autoregressive neural machine translation. ICLR 2018
>
> [6] M. Ghazvininejad, O. Levy, Y. Liu, and L. Zettlemoyer, “Maskpredict: Parallel decoding of conditional masked language models,” in EMNLP-IJCNLP, 2019, pp. 6112–6121.
>
> [7] J. Gu, C. Wang, and J. Zhao, “Levenshtein transformer,” NeurIPS, vol. 32, pp. 11 181–11 191, 2019
>
> [8] Gu, J. and Kong, X. Fully Non-autoregressive Neural Machine Translation: Tricks of the Trade. ACL 2021.

---

> > ### Comment · Reviewer_tzty · 2023-11-20
> > **Thank you for your response**
> >
> > After reading your response, Q1 is addressed. I raised my score from 3 to 5. However, I still have concerns in the proposed structure.  The novelty is quite limited to me, since I have read hierachical Transformer structure many years ago. I believe its efficiency is better, but it is not a fresh news, which makes the paper likes applying a method to another task.

---

> > > ### Author Response · Authors · 2023-11-20
> > >
> > > Thank you for your feedback and for increasing the score. In response to the concerns raised regarding the novelty of our work, we aim to underscore the distinctiveness of our approach within the context of utilizing the hierarchical transformer for speech generation. While the hierarchical transformer architecture has been previously explored, our research stands out due to its pioneering application in the domain of speech generation. We emphasize several critical innovations that significantly differentiate our work:
> > > 1. Specialization for Speech Generation: Our adaptation involves a specialized configuration of the hierarchical transformer expressly tailored for speech generation tasks. Unlike prior applications that focused on different domains, our model is purpose-built and optimized for speech generation.
> > > 2. Integrated Elements: We introduce several novel components within the architecture, including the incorporation of semantic tokens, acoustic tokens, in-context learning, Hi-Res audio, and multilingual considerations. This holistic integration of diverse elements fine-tunes the model's performance specifically for speech generation tasks, which is distinct from earlier implementations, making the proposed model a general speech language model.
> > >
> > > We believe these novel contributions collectively underscore the uniqueness and significance of our work in the field of speech generation. By specifically tailoring the hierarchical transformer for this purpose and introducing novel elements, our approach represents a substantial advancement in the realm of speech generation.

---

### Official Review · Reviewer_BzgH · 2023-10-30

**Soundness:** 3 good
**Presentation:** 3 good
**Contribution:** 2 fair
**Rating:** 6
**Confidence:** 4

**Summary:**

The paper presents a new approach to speech language modeling for producing both semantic and acoustic continuations of text/audio prompts, with the ability to retain speaker-specific voice characteristics. Previous approaches (AudioLM, VALLE) have relied on multi-stage chaining of Transformer models to model semantic vs. fine-grained acoustic characteristics of the generated continuations. This paper proposed a single-stage hierarchical Transformer architecture to accomplish both in a single pass.  The approach is evaluated on Librispeech clean test set and Aishell-2 (Chinese) with respect to word error rate and speaker similarity.

**Strengths:**

The paper advances current research on generative speech LMs in that it proposes a one-stage architecture to achieve both semantic continuity and fine-grained acoustic generation quality, where the cascading of representation generation at different levels of granularity is absorbed into the Transformer architecture itself. This is novel and will be of interest to audiences interested in audio generation/multimodality. From a theoretical perspective the proposed architectures delivers efficiency improvements.

**Weaknesses:**

1. The evaluations conducted raise several questions. Some of these may be due to a lack of clarity in the presentation and are listed below under Questions.
2. In addition to the theoretical efficiency analysis, it would be good to see actual measures of speed or latency in Table 3. The real-time efficiency depends on the number of layers among other factors, so it would be good to see this as part of the ablation study.

**Questions:**

For the semantic-to-acoustic condition in Table 1, why use WER as the evaluation criterion rather than human evaluation of acoustic generation quality? Does this box refer to the unconditional generation? If so, why not compare WER against the human-labeled ground truth, why were Hubert-Large/XLSR-53 used as 'Ground Truth'? The evaluation methodology/terminology is a bit confusing here.

---

> ### Author Response · Authors · 2023-11-15
>
> We sincerely appreciate the valuable comments and constructive suggestions to help improve the paper. We give the response below.
>
> ## Q1: The real-time efficiency.
>
> We provide our generation speed test with the fairseq-generate tool. The inference is run on the NVIDIA 3090 GPU in the semantic to acoustic mode with a batch size 64.
>
> |Model | $N_g + N_l$ | speed | # codec | # params|
> |----|----|----|----|----|
> |GPST | 11+4 | 2.31 sentences/s, 9438.75 tokens/s | 8 | 190M |
> |GPST | 10+8 | 1.89 sentences/s, 7643.03 tokens/s | 8 | 190M |
> |GPST | 9+12 | 1.57 sentences/s, 6232.50 tokens/s | 8 | 190M |
> |GPST-Hi-Res | 9+12 | 0.88 sentences/s, 7186.19 tokens/s | 16 | 207M |
>
> ## Q2: The reason for choosing WER as the evaluation metric in the semantic-to-acoustic task in Table 1.
>
> WER aims to evaluate the speech synthesis robustness. Neural TTS systems usually suffer from the robustness issue, which sometimes has deletion, insertion, and replacement errors due to wrong attention alignments. We perform ASR on the generated audio and calculate the word error rate (WER) with respect to the original text transcriptions. We have other metrics SPK and DNSMOS to evaluate the speech quality. These metrics focus on different perspectives on the speech generation models. It is hard for us to conduct human evaluation benchmarks since the baselines AudioLM and VALL-E are not open-sourced, we cannot perform a fair and rigorous MOS test for them.
>
> For the semantic-to-acoustic condition in Table 1, all the models except GPST-TTS are conditioned on the golden semantic tokens to generate acoustic tokens/waveforms, followed by an ASR model to calculate WER with respect to the text transcriptions. Only our GPST-TTS model can condition the text inputs to generate speeches. The GPST-TTS model first infers semantic tokens with texts as input, then generates acoustic tokens with the semantic tokens as inputs.
>
> ## Q3: The meaning of GroundTruth in Table 1 row 1.
>
> The results are directly calculated on the ground truth speech in the LibriSpeech test-clean dataset as the upper bound.

---

> ### Author Response · Authors · 2023-11-21
> **We are looking forward to your further comments.**
>
> Dear Reviewer BzgH,
>
> Thank you again for your insightful feedback on our submission. These valuable suggestions better strengthen the quality of our work. The deadline for the discussion stage is approaching, and we are looking forward to your further comments. If we have properly addressed your concerns, we would deeply appreciate it if you could kindly re-evaluate our paper. If you have further concerns. please let us know and we remain open and would be more than happy to actively discuss them with you.
>
> Best,
>
> Authors

---

> > ### Comment · Reviewer_BzgH · 2023-11-21
> >
> > Thank you, the main points of my review have been addressed. Please clarify them in the paper if it gets accepted, as other reviewers seem to have similar comments.

---

> ### Author Response · Authors · 2023-11-22
>
> Dear Reviewer BzgH,
>
> Thank you for your review and for acknowledging the efforts made to address the concerns raised in your evaluation. We appreciate your time and insightful feedback.
>
> While we are pleased that the responses have clarified the points you highlighted, we also note that the score remains unchanged. We understand that you have reservations about altering the score at this stage, and we respect your decision. However, we would like to kindly request your reconsideration of the assigned score.
>
> Should our paper be accepted, we are fully committed to implementing any further clarifications or modifications necessary to accommodate the suggestions provided, as you suggested.
>
> We value your expertise and insights and would greatly appreciate a reconsideration of the score in light of the addressed concerns. Your consideration in this matter is highly valued and will contribute significantly to the final assessment of our submission.
>
> Thank you for your time and consideration.
>
> Warm regards,

---

### Official Review · Reviewer_7zWQ · 2023-10-31

**Soundness:** 2 fair
**Presentation:** 3 good
**Contribution:** 3 good
**Rating:** 5
**Confidence:** 4

**Summary:**

* This paper studies the problem of speech language modeling. In particular, following the setup of SPEAR-TTS and AudioLM, speech is represented hierarchically as semantic tokens (clustered HuBERT/wav2vec2/w2v-BERT features) and acoustic tokens (SoundStream/Encodec tokens). A model first predicts semantic tokens and conditions on it to predict acoustic tokens.
* As acoustic tokens are derived from residual vector quantizers, each time step is represented as a stack of tokens and hence speech is typically encoded at a rate of 600Hz to 1200Hz. Modeling such long sequences can be challenging for vanilla language models. Prior work considers multi-stage models that model semantic tokens, acoustic tokens sequentially with separate models, which the authors argue would result in error propagation.
* The authors present a transformer to jointly model semantic tokens and audio tokens. In particular, MegaByte-style hierarchical Transformer is adopted for acoustic token modeling, where a global transformer processes sequence at 75Hz (taking the summed embedding over the stack of codes for a single time step) and the local transformer predicts the stack of codes within each time step auto-regressively.
* The proposed model evaluated quantitatively on semantic token-to-audio generation, voice conversion, and acoustic continuation tasks. Qualitative results on unconditional speech generation are presented in the appendix.

**Strengths:**

1. The authors presented a single model trained to optimize the joint probability of semantic tokens and acoustic tokens. In particular, the hierarchical Transformer facilitates efficient modeling of long sequences.
2. Empirical results demonstrate the effectiveness of the proposed method, outperforming AudioLM which is the most relevant work.

**Weaknesses:**

1. I don’t think it is fair to compare GPST/AudioLM with YourTTS/SPEAR-TTS/VALL-E on speaker identity transfer. The latter three are zero-shot text-to-speech synthesis models which take text and speaker prompt as input. In contrast, GPST and AudioLM take semantic tokens inferred from speech, which can have speaker information leaked from speech. If the authors follow the VALL-E setup for speaker identity transfer, then the semantic tokens are inferred from utterances from the same speaker as the audio prompt. For a more fair comparison, I would like to see the GPST-TTS setup for speaker identity transfer where semantic tokens are also inferred from text, in order to compare fairly with the zero-shot TTS models.
2. There is no direct comparison between hierarchical and non-hierarchical (i.e., flatten the audio tokens and use vanilla LM) models that are trained with the same objective. The comparison with AudioLM is not sufficient as there are other confounding factors.
3. Why are DNSMOS not presented in Semantic to Acoustic and Acoustic Continuation tasks?
4. GPST-Hi-Res lags behind GPST in most of the metrics. It is unclear why intelligibility (WER) and speaker similarity (SIM) degrades when having better audio quality

**Questions:**

1. What is the vocab size for E_a? Is it 1024 or 8192 (1024*8). Namely, does each residual codebook have its own embedding?
2. Is E_a shared between local and global transformers?
3. Unclear what local drops means. Each one predicts D codes for a time step and the batch size is b x T2. Is some (h_t, a_t^1, …, a_t^D) dropped to reduce batch size? Does it improve performance or just save memory?
4. How many layers for Ng, Nl? I did not find it in the paper or appendix

See weaknesses for other questions

---

> ### Author Response · Authors · 2023-11-15
> **Response part 1/2**
>
> We sincerely appreciate the valuable comments and constructive suggestions to help improve the paper. We give the response below.
>
> ## Q1: The evaluation of speaker identity transfer for GPST-TTS.
>
> We must clarify that our primary focus lies in developing a unified speech language model, while GPST-TTS is only a toy model for inspiration trained on LibriSpeech 960h, which is not comparable with VALL-E trained on Librilight 60k and SPEAR-TTS employing pre-training and back-translation techniques. However, we are still interested in the speaker information leak problem and give the extra experiment results here. Unlike VALL-E that directly trains with text transcriptions. If AudioLM and GPST use the semantic tokens inferred by a text-to-semantic model, it will cause a distribution mismatch with the semantic prompt provided in the speaker identity transfer stage. Because, as you state, semantic tokens may contain subtle speaker-related information such as prosody. Therefore, we propose a more rigorous inference mode for the text-to-semantic task like in-context learning in speaker identity transfer mode. We transcribe the speech prompt with an ASR model and prepend it to the text inputs in the text-to-semantic models, which is similar to the speaker identity transfer. We can see that GPST can get a significant improvement with text prompts from $0.547$ to $0.568$. We believe the performance can get further improvement if trained on Librilight 60k and using the pre-training and back-translation method proposed in SPEAR-TTS.
>
> In addition, we find that the semantic tokens can not control the speaker identity in the synthesized speeches. For example, a semantic sequence said by a man would still synthesize the speech in a woman voice. We provide some cases in the anonymous demo page (https://gpsticlr.github.io/GPST/demo).
>
> |Model | SPK | # codec | # params|
> |-----|-----|-----|-----|
> |**Speaker Identity Transfer**|
> |GPST-TTS w/o text prompt | 0.547 | 8 | 182M+190M|
> |GPST-TTS w/ text prompt | 0.568 | 8 | 182M+190M|
>
> ## Q2: Experiments on the non-hierarchical (flattened) vanilla LM.
>
> It's a natural thought to train a vanilla decoder-only transformer for speech generation. However, it is impractical since the length of the flattened acoustic tokens is too large (6000 tokens for 10s speech) to feed into the vanilla LM and will cause a cuda OOM error. This limitation stands as one of the key motivations driving the development of GPST, aiming to address and overcome these computational constraints in speech generation models.
>
> ## Q3: Why are DNSMOS not presented in Semantic to Acoustic and Acoustic Continuation tasks?
>
> Since both of AudioLM and VALL-E are not open-sourced, we follow AudioLM's instructions and compute DNSMOS with the examples provided on VALL-E’s demo page for fairness, which is described in Section 4.1.4. Unfortunately, VALL-E did not conduct the Semantic to Acoustic task and AudioLM did not conduct the Acoustic Continuation task. Therefore, the Speaker Identity Transfer task is the only one we can test to compare all the models in fair.
>
> ## Q4: GPST-Hi-Res has better audio quality but degrades in WER and SIM.
>
> The better audio quality means richer harmonic energy in the high-frequency regions, as shown in Figure 3, which focuses on different perspectives of the speech from WER and SIM. WER focuses on the robustness of speech models, which sometimes has deletion, insertion, and replacement errors due to wrong attention alignments. SIM focuses on speaker similarity to evaluate the voice cloning ability. Better speech quality does not guarantee better WER and SIM, which is the reason we choose three different metrics to evaluate the generated speech. As we mentioned in Section 4.2, the Hi-Res speech generation, with much more quantizers (from 8 quantizers to 16 quantizers), is still a tough task for speech-language models and requires a larger model with more parameters and longer training time. In the experiments, we simply use the same setting as non-Hi-Res models, which is not the best setting for Hi-Res model. Additionally, our model did not reach convergence before cessation of training. We believe the results would get improvement with more training steps.
>
> ## Q5: Does each residual codebook have its own embedding?
>
> Yes, each codebook has its own embedding and the vocab size for $E_a$ is 8192.
>
> ## Q6: Is $E_a$ shared between local and global transformers?
>
> No. Since the embedding sizes of the local transformer (512) and global transformer (1024) are different, we do not share the embedding table.

---

> ### Author Response · Authors · 2023-11-15
> **Response part 2/2**
>
> ## Q7: The clarification of Local-drop.
>
> The Local-drop method is proposed to save the GPU memory, without which the GPST-Hi-Res model would cause a cuda OOM error. The local transformer's input size is $(b \times T_2, D)$, so we can simply drop some samples in the batch dimension $b \times T_2$ since they are i.i.d. for the local transformer. The Local-drop method is only effective for hierarchical transformer because the unfolding operation in $T_2$ breaks the dependency in tokens to i.i.d., and is safe to drop them to save the memory. However, if we drop some tokens in the non-hierarchical (flattened) vanilla LM, we can not save memory since the sequence length and batch size are not changed.
>
> ## Q8: The settings of $N_g, N_l$.
>
> We ablate different settings of $N_g$ and $N_l$ in Table 3 with the number of total parameters fixed. In Table 1, we use the setting $N_g=9, N_l=12$ for evaluation. We will clarify it in the experimental setting.

---

> ### Author Response · Authors · 2023-11-21
> **We are looking forward to your further comments.**
>
> Dear Reviewer 7zWQ,
>
> Thank you again for your insightful feedback on our submission. These valuable suggestions better strengthen the quality of our work. The deadline for the discussion stage is approaching, and we are looking forward to your further comments. If we have properly addressed your concerns, we would deeply appreciate it if you could kindly re-evaluate our paper. If you have further concerns. please let us know and we remain open and would be more than happy to actively discuss them with you.
>
> Best,
>
> Authors

---

> > ### Comment · Reviewer_7zWQ · 2023-11-23
> > **Thank you for the response**
> >
> > I thank the authors for the response
> >
> > > We must clarify that our primary focus lies in developing a unified speech language model, while GPST-TTS is only a toy model for inspiration trained on LibriSpeech 960h, which is not comparable with VALL-E trained on Librilight 60k and SPEAR-TTS employing pre-training and back-translation techniques
> >
> > I understand the motivation and the goal. However, when you put different models into the same table for comparison, it should still be clearly stated whether the comparison is fair and what conclusions can/cannot be drawn. My earlier comment is that “you are comparing models taking different input representations and they are not fair comparison especially with potential speaker leakage issue when inferring semantic units from speech”
> >
> > > Unlike VALL-E that directly trains with text transcriptions. If AudioLM and GPST use the semantic tokens inferred by a text-to-semantic model, it will cause a distribution mismatch with the semantic prompt provided in the speaker identity transfer stage
> >
> > The claim of distribution mismatch is not clear to me. Models like SPEAR-TTS also trains a text-to-semantic model followed by semantic-to-audio token models. Why does it not have the issue? If speaker information is encoded in semantic tokens, then it is just a matter that text-to-semantic token prediction would already sample what speaker to be generated. What you would need to do is prefixing the generation at each stage with the tokens inferred from the reference (both semantic or acoustic)
> >
> > > However, it is impractical since the length of the flattened acoustic tokens is too large (6000 tokens for 10s speech)
> >
> > Could you conduct ablation studies in a reduced setup, such as reduced batch size / model size / or even sequence length? I believe such comparison is important to understand the performance difference.
> >
> > In addition, another comment I made is about is the authors should compared single stage vs two stage model (like AudioLM) in a controlled setup. For this experiment, the second stage could still be a hierarchical one and should not suffer the OOM issue the authors mentioned. It is also important to understand if there is any performance benefit/regression using a single stage model. We cannot directly draw conclusion based on the comparison with AudioLM since AudioLM uses different codec so there are cofounding factors to the performance change.
> >
> > > DNSMOS
> >
> > I would still encourage the authors to add those numbers for your own models in those tasks to confirm if GPST/GPTS-Hi-Res gain is consistent as well as checking the performance of GPST-TTS
> >
> > I still consider there are open questions especially on the ablation studies and comparison with alternative methods pointed out by other reviewers (e.g., delay pattern). After reading the review carefully, I would still keep my original rating

---

### Official Review · Reviewer_3McF · 2023-11-04

**Soundness:** 3 good
**Presentation:** 3 good
**Contribution:** 2 fair
**Rating:** 5
**Confidence:** 5

**Summary:**

This paper studies the task of speech-language modeling. The authors proposed a hierarchical transformer approach to train a speech-based language model, denoted as Generative Pre-Trained Speech Language Model (GPST). GPST has both global and local transformers. The global transformer operates on "coarse" units while the local transformer operates over the "fine" units.

The authors compare the proposed method to several baseline methods considering content preservation (WER), speaker similarity (SPK), and overall quality (DNSMOS). The authors additionally present their method is capable of performing under the multi-lingual setup, and a small ablation study.

**Strengths:**

1. The authors study the task of speech-language modeling using a joint training of global and local models.
2. The local and global models mitigate the issue of long sequence generation which is a critical issue in speech LMs.
3. The authors empirically show the proposed method is superior to the evaluated baselines.

**Weaknesses:**

1. The global and local transformers were introduced before in the NLP community.
2. There are various unsupported claims in the paper.
3. Overall this paper seems incremental considering the other modeling approaches such as AudioGen, VALL-E, and AudioLM. It is not clear whether this is enough to pass the bar for ICLR publication.

**Questions:**

1. The global and local transformers were introduced before in the NLP community. This makes the novelty of the proposed approach rather limited as it seems like a different application of the same modeling approach. Can the authors provide more details on any adjustments needed for the model to be applied to the evaluated tasks?
2. There are various unsupported claims in the paper. For instance:  “...which limits the performance of fine acoustic token generation..“ why this is true? did the authors study that? can the authors provide experiments/refs to support that? Or: "These multi-stage generative models induce significant error propagation issues, which can negatively impact the overall performance.“ do the authors have experiments/refs to support that? Or: "Additionally, obstructing the information flow among hierarchical quantizers would degrade the model’s performance, especially in Hi-Res speech generation that requires more residual quantizers."
3. Regarding the efficiency analysis, I’m not sure I agree with the author's analysis. The size of $D$ is usually 4/8/12 not more than that, $m_l$ is probably much bigger. So in case the authors decide to ignore $m_l$ in their analysis, they should also ignore $D$. Overall, I agree with the authors that the proposed method is likely to be more efficient, I just do not agree with their specific analysis.
4. Regarding the results, can the authors provide more details on how they compute the WER using the GSLM method? GSLM [1] is a decoder-only method that operates on units obtained from HuBERT. There is no conditioning on text. So it is not clear to me how the authors compute WER when the model performs continuation / unconditional generation. Can the authors provide more details here?
5. Did the authors try to explore / compare their method to a simple delay pattern as introduce by [2, 3]? It will greatly simplify the modeling approach and remove the need to the local transformer.
6. If I'm not mistaken, the paper's length should be 9 pages, with an unlimited number of pages for refs. and supplemental material. Hence, I believe the "REPRODUCIBILITY AND ETHICS STATEMENT" statements provided by the authors on the 10th page break the submission guidelines.

I'm willing to change my score in case I'm mistaken.

[1] Lakhotia, Kushal, et al. "On generative spoken language modeling from raw audio." Transactions of the Association for Computational Linguistics 9 (2021): 1336-1354.
[2] Kharitonov, Eugene, et al. "Text-free prosody-aware generative spoken language modeling." arXiv preprint arXiv:2109.03264 (2021).
[3] Copet, Jade, et al. "Simple and Controllable Music Generation." arXiv preprint arXiv:2306.05284 (2023).

---

> ### Author Response · Authors · 2023-11-15
> **Response part 1/2**
>
> We sincerely appreciate the valuable comments and constructive suggestions to help improve the paper. We give the response below.
>
> ## Q1: The model architecture was introduced before in the NLP community and the novelty is limited compared to previous speech models.
>
> It is true that the hierarchical transformer is not a new architecture in the NLP community. However, our focus lies specifically on the speech generation task, where the utilization of the hierarchical transformer remains unprecedented. Our work pioneers the application of this architecture for speech generation, marking a significant departure from prior endeavors. Notably, our approach involves a novel design of the architecture, encompassing the integration of semantic tokens, acoustic tokens, in-context learning, multilingual aspects, among others, tailored specifically for the speech language model. This task distinctly differs from conventional text models, thereby underscoring its novelty.
>
> While acknowledging the foundational groundwork laid by AudioLM in the realm of speech generation, it is imperative to note that our model, GPST, draws inspiration from AudioLM while innovatively leveraging the hierarchical transformer to address inherent challenges encountered in AudioLM and VALL-E. These preceding models face significant limitations when generating lengthy acoustic sequences, a hurdle meticulously addressed in Section 3.1. Our model not only tackles these challenges but also outperforms them, as substantiated by our empirical findings.
>
> ## Q2: Unsupported claims in the paper.
>
> We will carefully clarify the unsupported claims as follows:
>
> ### (1). ``...which limits the performance of fine acoustic token generation..''
>
> We state it in the introduction of VALL-E. As described in the paper, VALL-E conducts non-auto-regressive generation of the acoustic tokens from subsequent quantizers. As investigated in non-auto-regressive generation models [1,2,3,4], the independence assumption in the output space ignores the real dependency between target tokens, and the maximum-likelihood training forces to cover all possible modes, which makes NAR systems perform worse than the AR baselines. VALL-E leverages the vanilla non-auto-regressive generation model for the subsequent acoustic tokens, typically resulting in suboptimal performance compared to auto-regressive generation models.
>
> ### (2). ``These multi-stage generative models induce significant error propagation issues, which can negatively impact the overall performance.''
>
> It is established in the literature that cascaded multi-stage systems often exhibit inferior performance in comparison to end-to-end systems due to inherent error propagation issues [5,6]. This phenomenon stems from the discrepancy between the training and inference procedures within cascaded systems. During training, all modules receive training signals derived from ground truth labels. However, during inference, these modules are fed outputs from preceding modules, resulting in error outputs that are out of distribution concerning the training data. Consequently, as elaborated in Section 3.1 and illustrated in Figure 1, both AudioLM and VALL-E adopt cascaded architectures, inherently predisposing them to the issue of error propagation. This discrepancy contributes to the amplification of errors throughout the cascaded system's feed-forward flow, thereby adversely affecting performance.
>
> ### (3). ``Additionally, obstructing the information flow among hierarchical quantizers would degrade the model’s performance, especially in Hi-Res speech generation that requires more residual quantizers.''
>
> The acoustic tokens are obtained with residual quantizers, wherein the subsequent quantizers rely heavily on the results from the previous quantizers. In the Hi-Res speech generation setting, the number of residual quantizers (n=16) is much larger than the normal setting (n=8), which poses more challenges for the model. With the two issues described above, the information flow in VALL-E is obstructed by non-auto-regressive model and cascaded systems, resulting in the degradation of the model's performance.
>
> We will add the support evidence and related citations to the revised version, thanks for helping improve our work.
>
> ## Q3: Regarding the efficiency analysis.
>
> The efficiency analysis comes from two different perspectives, one is computation complexity and the other is FLOPS. In the computation complexity analysis, if we omit $D$, which implies flattening the acoustic sequence, we will get a cuda OOM error on GPU. Therefore, we keep it in our analysis and ensure a comprehensive evaluation of the computational complexity.

---

> ### Author Response · Authors · 2023-11-15
> **Response part 2/2**
>
> ## Q4: The details of the WER computation in GSLM.
>
> GSLM uses HuBERT code (semantic tokens) as inputs like AudioLM and reconstructs the waveform with the Tacotron2 model and the WaveGlow vocoder. We run their open-sourced code using the released model and evaluate the results. GSLM can not generate speeches conditioned on texts, and the evaluation is done directly based on golden semantic tokens. In the Semantic to Acoustic part of Table 1, only our GPST-TTS model can generate speeches conditioned on texts.
>
> ## Q5: The comparison with delay pattern models.
>
> The delay pattern models in [7,8] are tailored for semantic tokens like GSLM and music generation. Designing a dedicated model for speech LM incorporating acoustic tokens in a delay pattern extends beyond the scope of our work. However, we provide an analysis here to illustrate why our model is better than the delay pattern approach. The delay pattern model predicts multiple tokens in parallel with a simple linear classification head, while GPST predicts them with a local transformer in an auto-regressive manner, which means GPST can better fit the distribution. The auto-regressive mechanism and the utilization of a local transformer within GPST afford it a superior capacity to model intricate dependencies among tokens, thereby establishing its theoretical superiority over the delay pattern approach.
>
> ## Q6: Regarding the length of the paper.
>
> As ICLR 2024 Author Guide states (https://iclr.cc/Conferences/2024/AuthorGuide), the optional ethic statement and reproducibility statement will not count toward the page limit, but should not be more than 1 page. So we believe we didn't break the submission guidelines.
>
> [1] Gu, J., Bradbury, J., Xiong, C., Li, V.O. and Socher, R., 2017. Non-autoregressive neural machine translation. ICLR 2018
>
> [2] M. Ghazvininejad, O. Levy, Y. Liu, and L. Zettlemoyer, “Maskpredict: Parallel decoding of conditional masked language models,” in EMNLP-IJCNLP, 2019, pp. 6112–6121.
>
> [3] J. Gu, C. Wang, and J. Zhao, “Levenshtein transformer,” NeurIPS, vol. 32, pp. 11 181–11 191, 2019
>
> [4] Gu, J. and Kong, X. Fully Non-autoregressive Neural Machine Translation: Tricks of the Trade. ACL 2021.
>
> [5] Inaguma, H., Popuri, S., Kulikov, I., Chen, P.J., Wang, C., Chung, Y.A., Tang, Y., Lee, A., Watanabe, S. and Pino, J., 2022. UnitY: Two-pass Direct Speech-to-speech Translation with Discrete Units. ACL 2022.
>
> [6] Barrault, Loïc, et al. "SeamlessM4T-Massively Multilingual \& Multimodal Machine Translation." arXiv preprint arXiv:2308.11596 (2023).
>
> [7] Kharitonov, Eugene, et al. "Text-free prosody-aware generative spoken language modeling." arXiv preprint arXiv:2109.03264 (2021).
>
> [8] Copet, Jade, et al. "Simple and Controllable Music Generation." arXiv preprint arXiv:2306.05284 (2023).

---

> > ### Comment · Reviewer_3McF · 2023-11-20
> > **Response to authors**
> >
> > I would like to thank the authors for their detailed responses and clarifications.
> > After carefully reading the rebuttal, I still believe the novelty of the proposed method is limited and the authors should compare their method to more relevant baselines such as a simple delay pattern.
> > Moreover, I still find the comparison to GSLM a bit strange as it does not measure any modeling, but only shows the benefit of using acoustic tokens on top of semantic tokens (which was already demonstrated in the AudioLM paper).
> >
> > Due to all of the above, I would like to keep my score unchanged.

---

> > > ### Author Response · Authors · 2023-11-20
> > >
> > > Thank you for the feedback.
> > >
> > > 1. Acknowledging the time constraints, we regret that reproducing the delay pattern model within the given timeframe is impractical due to the extensive training duration. We emphasize that the comparison with AudioLM and VALL-E have demonstrated the performance of our model.
> > >
> > > 2. It's important to clarify that selecting GSLM as a baseline for comparison is just a common experimental setting in prior studies such as AudioLM and VALL-E. Notably, our proposed model, GPST, has demonstrated superior performance compared to the AudioLM model, surpassing GSLM significantly. Consequently, the conclusions drawn in our paper remain unaffected by this comparison.

---

### Comment · Area_Chair_cTyd · 2023-11-10
**reviewer-author discussions**

Dear All,

The reviewer-author discussion period will be from Nov. 10 to Nov. 22. For reviewers, please read the authors' responses and acknowledge it, respond to them early on in the discussion, and discuss points of disagreement. Thank you!

AC

---

### Author Response · Authors · 2023-11-15
**Anonymous Demo Page**

Dear all,

We provide a demo page for experiencing the quality of generated speeches (https://gpsticlr.github.io/GPST/demo).

---

### Meta-Review · Area_Chair_cTyd · 2023-12-05

**Metareview:**

The paper proposes a novel generative pre-trained speech language model (GPST) that can generate high-quality speech. The paper claims that GPST can overcome the limitations of previous speech language models that rely on a multi-stage generation process and suffer from error propagation and information loss. As a solution, the paper proposes to use a single hierarchical Transformer which consists of a global transformer to contextualize representations and a local transformer to predict the stack of codes within each time step auto-regressively. The proposed method well addressed the challenge of long sequence modeling that previous speech language models have faced. The GPST method was compared with several popular methods such as GSLM, AudioLM, and VALL-E etc. to show its advance in the semantic-to-acoustic, the speaker identity transfer, and the acoustic continuations tasks.

Key strengths:

A single model is trained to optimize the semantic and acoustic tokens.

The solution helps to reduce the pain of modeling very long sequence codes, handling different tokens using the global and local transformers inside a single model.

Emipriccal results show the effectiveness of the proposed method in the tasks set up by the authors.


One major weakness of this paper is limited novelty, as it applies the hierarchical Transformer previously proposed for the NLP task to the speech generation task. Although the authors argued that they have lots of works (e.g., “including the incorporation of semantic tokens, acoustic tokens, in-context learning, Hi-Res audio, and multilingual considerations” to adapt the original hierarchical Transformer to fit the speech generation application, such adaptation becomes very specific to the speech generation task, which constrains the work from attracting large amounts of ICLR audience from different areas.

Furthermore, the experiment setups may be unfair, as pointed out by Reviewer 7zWQ. For example, the GPST model has the potential speech leakage to get the performance advantage over all the zero-shot TTS models. It is hard to reach the paper’s claim without a fair comparison. Reviewer 3McF especially have concerns about multiple unsupported claims in the paper.

Another note is about the NAR modeling which significantly improves the inference speed. For example, the combination of the AR model and the NAR model in VALL-E provides a good trade-off between speech quality and inference speed. It may not be good to simply dismiss NAR models with quality degradation.

We encourage the authors fully address the above concerns in their next version of the paper.

**Justification For Why Not Higher Score:**

There are too many concerns on this paper such as the novelty and experiment verification. The biggest concern is the proposed method just adapts the hierarchical Transformer previously proposed for the NLP task to the speech generation task. The model is an existing one, although the application is new. Compared to the ICLR 2024 submissions in the speech area, this paper ranks at the bottom.

**Justification For Why Not Lower Score:**

N/A

---

### Decision · Program_Chairs · 2024-01-16

Reject